# Attitudes Formation toward Minority Outgroups in Times of Global Crisis—The Role of Good and Bad Digital News Consumption

**DOI:** 10.3390/bs14030232

**Published:** 2024-03-13

**Authors:** Nonna Kushnirovich, Sabina Lissitsa

**Affiliations:** 1Ruppin Academic Center, Department of Economics and Management, Emek-Hefer 4025000, Israel; 2School of Communication, Ariel University, Ariel 407000, Israel; sabinal@bezeqint.net

**Keywords:** attitudes toward immigrants, ethnic minority, digital news consumption, threat, COVID-19

## Abstract

This paper examines the relationships between the consumption of ‘bad’ or ‘good’ digital economic news and attitudes toward immigrant and ethnic minorities during the crisis that developed during the COVID-19 pandemic. The study considered attitudes toward two minority groups in Israel: immigrant citizens from English-speaking countries, and Israeli Palestinian citizens, an ethnic minority. The data were collected through an online survey of 866 respondents, who were members of the majority population group. The study found that, during the global crisis, exposure to bad digital news was associated with more positive attitudes toward both disadvantaged and non-disadvantaged minority groups. Moreover, in times of global crisis, people focused mostly on local rather than global digital news. In contrast to the idea of Intergroup Threat Theory, the study revealed that feelings of economic threat during the global crisis engendered higher cohesion between different population groups, and more positive attitudes toward minorities. In times of crisis, bad news for the economy brings good news for social solidarity—people tend to rally around the flag; this phenomenon even occurs between groups engaged in years-long, protracted conflict.

## 1. Introduction

During times of crisis, people experience threats to their economic, social, and health resources. Because of the lockdowns during the COVID-19 pandemic, many shops and service businesses were closed, which led to an increase in unemployment and food prices, and a decrease in business revenues [1]. In crises, the main sources of information about the developing situation are the mass media, such as TV news and social media [2]. In fact, digital news consumption significantly increased during COVID-19 [3,4], earning the latest pandemic the sobriquet “infodemic” [5] due to the massive information overflow. More information usually means less uncertainty; however, evidence indicates that news consumption during a crisis actually increases feelings of both general and individual economic uncertainty and insecurity [6]. In such a situation, it is unclear how a flood of mostly bad digital news about the economic situation is reflected in other spheres of social life, such as in intercultural relations, which is the focus of the current study.

In the first months of the pandemic, the public’s response reflected solidarity and empathy [7]. Many governments stressed that “we are all in this together”, arguing that people could only survive the outbreak by manifesting solidarity. This message resonated with the public, and measures to help vulnerable groups were launched. Moreover, because this was a global pandemic, people initially expressed solidarity with others across the globe, especially with disadvantaged countries that suffered more severely from the pandemic and its consequences [7]. However, as the pandemic continued, intergroup solidarity was challenged, new forms of stigma developed, and racism and xenophobia increased [8]. The outbreak of a pandemic tends to reveal existing societal prejudices, which are driven by the fear of infectious disease and a desire to blame some “other”, who is perceived as “responsible” for the pandemic [9]. During the pandemic, a type of politicized ethno-cultural racism re-emerged, especially toward people from Chinese backgrounds, even in countries and areas whose governments declared “we are in the same boat” policies, such as in North America [10], the UK [11], and Germany [12]. The prejudice and discrimination expanded to include those who had never visited their ancestral lands or were from other Asian countries [13]. Although numerous studies have addressed racism, xenophobia, and the stigmatization of Asian minorities [9,10,11,13], which were fully associated with the illness, the attitudes toward other minorities that competed with the majority population for the scarce economic resources during the economic recession have been almost totally overlooked. 

Numerous studies have explored the relationships between the threats posed by minorities and attitudes toward them [14,15,16]. Some studies have investigated the impact of threats to objective economic conditions (for a review, see [17,18]); others have focused on subjective perceptions of economic conditions and attitudes toward immigrants [18,19,20]. However, the focus of most of these studies has been on attitudes toward disadvantaged groups of immigrants, without considering the sources of such perceptions or whether they are associated with digital news consumption. 

Another corpus of studies investigating news consumption and attitudes toward minorities focused on news about minorities themselves and not on economic digital news [21,22,23], although some of these studies addressed the importance of digital news in general. The impact of digital news coverage of ethnic and immigrant minorities on attitudes toward them depends on the context in which such news was received [20]. In times of crisis, the effects of news may be more strongly pronounced or even different from those during periods of greater stability. Moreover, evaluating the content also matters. Boomgaarden and Vliegenthart [24] found that qualitative characteristics—positive or negative evaluations of immigrants in the news—were a stronger predictor of attitudes toward immigrants than quantitative characteristics, i.e., frequency of mention. However, the role of the consumption of positive (‘good’) and negative (‘bad’) economic digital news during the pandemic, regarding both local and global economic events, in the development of intergroup attitudes remains overlooked. 

Generally, it is very difficult to describe an economic news item as inherently good or bad because sometimes news is politically oriented, and news evaluation depends on individual preferences. A story about a government that, for example, chooses to dramatically increase welfare support for certain groups (which happened in various countries during COVID-19) could be framed as good economic news for one group, but perhaps as bad news for another group. In order to categorize economic news as negative (‘bad’) or positive (‘good’), it is recommended to focus on the news tone, which was defined by Kleinnijenhuis et al. [25] through the use of terms associated with dread and hope. News using words associated with optimism, assurance, enthusiasm, encouragement, relief, grip, rescue, and revival is gauged to have a positive tone and is perceived as ‘good’ news. News using words associated with worry, shock, fear, danger, concern, chaos, tension, nervousness, and anxiety may be regarded as negative (‘bad’) news [26]. Thus, economic news cannot be defined a priori as good or bad; this depends on people’s subjective perceptions of whether the news they consume is good or bad.

Given that the crisis caused by the pandemic was initially a health crisis, and only then an economic crisis, studies examining interactions between media consumption and threats during the COVID-19 pandemic mostly considered health threats or threat as perceived by government policy, but not pure economic threat [27]. The rare studies on economic threat caused by the pandemic found that economic threat functioned differently from health threat [28,29]. However, they did not relate to the impact of economic digital news consumption on attitudes toward minorities during the pandemic, which was not a ‘classic’ economic crisis but a global one. To the best of our knowledge, no studies have investigated how the consumption of news about the economic situation in the national economy and abroad during the COVID-19 pandemic shaped attitudes toward minorities, while distinguishing between non-disadvantaged and disadvantaged minority groups. This research aims to fill this void.

The purpose of this study was to examine how exposure to bad or good digital news about the economic situation in the local national economy or the economy abroad during COVID-19 explained attitudes toward minorities. We examined the attitudes toward two different minorities in Israel that hold Israeli citizenship: an ethnic minority—Israeli Palestinian citizens (a disadvantaged minority with whom the Jewish Israeli majority is in protracted conflict); and an immigrant minority—immigrant citizens from English-speaking countries (non-disadvantaged minority group). 

The study examined the relationships between economic digital news consumption and attitudes toward immigrant and ethnic minorities in the midst of the crisis that emerged during the COVID-19 pandemic. Because the study focused on economic aspects of the crisis, and as most crises are usually accompanied by economic downturn, the results may be useful for understanding what happens to attitudes toward minorities during any times of global crisis.

### 1.1. The Israeli Context

*Immigrant minority—immigrants from English-speaking countries in Israel*. Israel is a nation of Jewish immigrants. They are eligible for Israeli citizenship under the Law of Return, which grants Jews and their descendants immediate citizenship. Immigrants from English-speaking countries are a high-skilled [30] and high-income group [31,32]; about 65 percent of them have an academic education, as compared to about 48 percent of the native-born Israeli population. The discrepancy in lifestyle between the country of origin and the destination country can often be striking for newcomers, but this is not the case for English-speaking immigrants (ESIs) in Israel, for whom shared cultural values and the generally positive attitudes of Israelis toward the US help to ameliorate some of the dissonance of the immigration experience [33]. Unlike other high-skilled immigrant groups, English speakers’ linguistic and professional skills are more easily transferable to the Israeli job market [32], which makes their economic integration more successful [34]. Many English-speaking immigrants settle in English-speaking communities, enhancing the American culture bubble; however, compared to other minorities, they report extremely low levels of feeling discriminated against [34]. 

*Ethnic minority—Israeli Palestinians*. In national statistics, the Israeli population is typically broken down into Jews (approximately 75%) and Israeli Palestinians (IPs) (about 20%). Despite their Israeli citizenship, the IP minority is a separate minority group with a different culture, religion, and language from the Jewish majority. Even though they are Israeli citizens, IPs face social isolation. Due to existing social stratification, IPs are at a disadvantage compared to Jews in a variety of areas, including occupational status, income, education, standard of living, and access to health facilities [35]. The IPs’ income is lower than that of Jewish Israelis and immigrants from English-speaking countries; they also report a high level of discrimination in Israeli society [34]. Recent studies have shown that Israeli Jews harbor deep-seated antipathy, resentment, and prejudice toward Israeli Palestinians [36]. 

Israel is a proper case for such a study because Israel’s media consumption is characterized by a profusion of media sources, websites, and digital content [37].

### 1.2. Theoretical Background and Research Hypotheses

#### 1.2.1. Exposure to Economic Digital News and Perceived Economic Threat

In times of global crises like the COVID-19 pandemic, people feel the threat of losing personal economic resources, such as job loss, reduction in income, mounting debt, as well as worsening national economic situation, reflected in inflation, local and global economic instability, or even socio-political unrest [29]. Intergroup Threat Theory (ITT) [38] states that outgroups may represent both a symbolic threat (threat to ideals, culture, and values) and realistic threat (threat to the group’s resources, power, and security). Realistic threats also include an economic threat to the ingroup’s wealth. ITT conceptualizes an economic threat as a dimension of realistic threat. Economic threat revolves around the perceived rivalry for limited resources, such as employment opportunities, housing, and social benefits, and the belief that resources designated for local residents are jeopardized by other groups [38,39,40]. A study conducted during the COVID-19 pandemic in eight high- and middle-income countries showed that people felt threatened with economic loss for themselves, their country, and the world [41]. Even the idea that an outgroup threatens the ingroup can have detrimental psychological effects [38].

Information about possible losses and threats is disseminated by the media. Jacobs et al. [42] found that, in the Netherlands, exposure to TV news was associated with more negative emotions regarding the economy (namely, feeling a threat to the economy), which in turn impacted Dutch citizens’ attitudes toward Muslims. During the COVID-19 pandemic, social media users who believed that content on social media was fair were more likely to trust that Chinese people posed a realistic threat to America [43].

The subjective nature of perceiving economic news as either positive or negative underscores the complexity of interpreting such information [44]. Moreover, media audiences usually regard positive and negative information differentially, a phenomenon known as negativity or positivity bias or valence asymmetry. Generally, negative information is remembered more and given greater weight than positive information [45]. Heavy media consumption of “bad news” fosters inaccurate or exaggerated notions of victimization, mistrust, and danger [46,47], which may heighten perceived threat. For example, in explaining the effects of continuous exposure to news on economic recessions as a macro-level event, Pearlin and Bierman [48] argued that, when economic troubles are frequently covered in the media, the misfortunes of others can cause anxiety among individuals who have not personally experienced such hardship. Increased exposure to news about consequences of the COVID-19 pandemic, including economic consequences such as business closures and job losses, heightened perceptions of threats to economic security [49].

Another research stream emphasized the effectiveness of positive messaging in a variety of circumstances [50]. People process positive information faster, have broader associations with it, and seem to draw broader inferences from it compared to negative information [45]. According to an increasing body of research, people update their expectations in an asymmetrical manner: they tend to ignore unfavorable information while more readily considering favorable information [51,52]. Taking the “newsworthiness criteria” into account, we may assume that if, during the era of the pandemic, with its insecurity and need for constant updates, good economic digital news nonetheless passes the media gatekeepers and is broadcast, it must presumably be dramatic, influential, and optimism-evoking. For example, the story of a business that found new ways to generate revenues during the pandemic may be perceived as inspiring hope for an exit from the recession, and therefore reduce perceived economic threat. Similarly, optimistic news on other countries’ success in overcoming the economic crisis may be domesticated and provide hope for an exit from economic recession in the audience’s own country as well, diminishing the perception of economic threat for local media consumers. 

Scholars distinguished between different kinds of media in terms of their effects on people’s perceptions during the COVID-19 pandemic. For example, Dhanani and Franz [53] found that those who used news websites reported fewer negative attitudes toward Asian Americans, but that consumers of social media and print news sources had more negative attitudes. However, such differences may be related to the fact that people with higher levels of prejudice choose certain kinds of media. For instance, Cho et al. [54] found that using social media and TV Fox News for COVID-19 news predicted stigmatization, whereas no such relationship was found for CNN news. Moreover, different news outlets are likely to publish content with different valency: some of them tend to post negative content, and others post more positive content [55]. Thus, it is not certain kinds of media but rather news valence that plays a role in this process.

The localization of news also matters. During a global crisis such as the COVID-19 pandemic, the local media constantly expose the audience to global news, creating the impression of economic collapse everywhere. “Far away” events resonate and become newsworthy for a local media audience due to domestication mechanisms, which, by discursively adapting news from outside a country, make them comprehensible, appealing, and more relevant to domestic audiences [56]. The audience localizes overseas news, and perceives foreign events as a ‘mirror’ of similar local experiences [57]. For instance, a financial crisis caused by COVID-19 in a faraway country may become part of domestic discourse regarding economic and financial issues, and evoke anxiety and threat in a way similar to local economic “bad” news. Accordingly, we expressed the following hypotheses: 

**H1.** 
*The frequency of exposure to bad digital news about the economic situation in the national economy (H1.1) and abroad (H1.2) will be positively associated with economic threat.*


**H2.** 
*The frequency of exposure to good digital news about the economic situation in the national economy (H2.1) and abroad (H2.2) will be negatively associated with economic threat.*


#### 1.2.2. Perceived Economic Threat and Attitudes toward Minorities

Intergroup anxiety is frequently encountered by individuals, even without engaging in social interactions with the outgroup. It frequently occurs between people from various cultural, racial, and ethnic backgrounds, as well as between people from stigmatized and non-stigmatized groups. It stems from the fear of unfavorable outcomes and feelings of threat [40].

Ethnic competition theory [58] stresses the role of economic competition among ethnic groups. According to this theory, higher economic threat is associated with heightened intergroup tensions and conflicts between advantaged and disadvantaged ethnic groups since these groups compete for limited resources [59]. The COVID-19 pandemic posed a new viral threat that exacerbated anti-immigrant sentiments. The economic threat experienced during the pandemic was commonly followed by anxiety and feeling threatened [29]. During a crisis, when resources constraints are felt more strongly and when competition for resources is intensifying, prejudice may increase and attitudes toward outgroups may worsen. Semyonov et al. [20] claimed that attitudes toward immigrants became more negative when the competitive threat grew. Other studies also found that macro-economic threats increased prejudice against ethnic and immigrant groups [60,61]. People who viewed immigrants as less of an economic threat were more likely to provide immigrants with empowerment assistance [62]. COVID-19 revealed and aggravated global prejudice and inequality [63]. Dhanani and Franz [53] found that people who focused on the economic threat of the COVID-19 virus showed higher levels of prejudice against Asian people, although those who focused on the health threat did not. Because of the lockdowns, young people, business owners, and even middle-class employees who were unable to work remotely entered the circle of poverty. Many of them were forced to work in less prestigious jobs, rely on unemployment benefits, and compete with disadvantaged minorities for social welfare. Disadvantaged population groups that were even vulnerable socially, economically, and in terms of health pre-pandemic were further alienated and marginalized by COVID-19 [64]. In line with ethnic competition theory, such an acceleration of social polarization may severely undermine social cohesion and aggravate already tense intercultural relations [65]. This may especially be the case in Israel regarding the relationship between the native-born Jewish majority and ethnic and immigrant minorities. The size of the minority population (about 20% of the Israeli population are ethnic minority Israeli Palestinians, and about 21% are immigrants) and the economic crisis (in our case, caused by lockdowns) are seen as two key sources of competition over scarce economic resources [19]. Accordingly, we expressed the following hypotheses:

**H3.1.** 
*Economic threat will be associated with less positive attitudes toward Israeli Palestinian citizens.*


**H3.2.** 
*Economic threat will be associated with less positive attitudes toward English-speaking immigrants.*


#### 1.2.3. Exposure to Economic Digital News and Attitudes toward Minorities

The attitudes of the majority population toward immigrant minorities are not consistent over time. They fluctuate due to societal and economic ‘external shocks’, such as increasing unemployment or immigrant waves. This is because, in these periods, people are more susceptible to news information [20]. Kuntz et al. [19] found that anti-immigrant sentiments were lower in countries with lower perceptions of economic insecurity, and that these perceptions were better predictors of attitudes toward immigrants than changes in objective economic conditions. Subjective perceptions about the economic situation are commonly shaped by the news [18]. Thus, consuming news about the economic situation may play an important role in shaping attitudes toward immigrant and ethnic minorities [19]. In line with this, Sorokowski et al. [66] found a positive relationship between media exposure to information about the COVID-19 pandemic crisis, its consequences, and prejudice towards foreigners. However, some studies revealed that, during the COVID-19 pandemic, intensive consumption of digital media news and higher perceptions of threat were not necessarily associated with more negative attitudes toward minorities. For example, Crouche et al. [43] found that, during the COVID-19 pandemic, social media consumers reported higher realistic threat perceptions but lower intergroup anxiety.

Uncertainty–identity theory explains how feelings of uncertainty inspire individuals to identify with social groups, to join new groups, or reconfigure current ones in order to reduce, control, or protect from feelings of uncertainty. One of the most popular strategies adopted when striving to resolve, manage, or avoid feelings of uncertainty is the appeal to group identification. Group identification gives us a feeling of who we are and how we should feel and think, and also lessens our confusion over how other people will respond and the direction that social engagement will take. People “join” new groups, identify or identify more strongly with existing self-inclusive categories (such as one’s nation), or attach themselves to groups that they already “belong to” (such as one’s work team) when they are unsure of themselves or things that reflect themselves [67]. 

In a crisis, uncertainty is anxiety-provoking and distressing since it makes people feel as though they cannot predict or control what will happen to them [67]. The COVID-19 pandemic was a macro-level stressor that exacerbated overall uncertainty at a national level [49,68] owing to lack of information about its duration and its impact on the world and on the local and global economy [69]. The widespread exposure to news media via 24 h news cycles, providing mostly inconsistent details about the pandemic and its economic and social consequences, in fact increased uncertainty instead of reducing it [49]. In line with the uncertainty–identity theory, we may assume that, in order to reduce the uncertainty, Israeli Jews may act in the following ways: (1) identify more strongly with their own group, distinguishing themselves from newcomers (ESIs) and minorities (IPs); (2) reconfigure their ingroup, including the newcomers (ESIs) within it on the basis of their common ethnic and religious identity; (3) reconfigure their ingroup, including IPs within it on the basis of their being Israeli-born, speaking Hebrew, being familiar with Israeli culture, etc.; and (4) identify more strongly with an existing self-inclusive category—Israelis—and accordingly count all Israeli citizens as part of their broad ingroup, including ESIs and IPs. 

Based on uncertainty–identity theory and existing research demonstrating that intensive news consumption increases uncertainty [49], and in light of the ambiguity concerning how news consumption relates to attitudes toward minorities during a crisis, we formulated the following research question:

RQ1. What will be the relationships between frequency of exposure to bad/good economic digital news during the global crisis, and attitudes toward ESIs and IPs?

The conceptual model of the study is presented in Figure 1.

Due to the specific character of the COVID-19 pandemic crisis, the perceived health threat was also an important factor. Moreover, actual losses of health and economic resources might shape feelings [29]. Accordingly, the research model examined in this study controlled for health threat, being harmed economically during the COVID-19 pandemic, health impairment during the COVID-19 pandemic, income, and demographic characteristics (age, gender, education). 

## 2. Materials and Methods 

### 2.1. Procedure

This research used data from a November 2020 online survey of Israeli Jewish digital news consumers aged 18+. By this time, two waves of COVID-19 and two very restrictive and lengthy lockdowns had passed in Israel and the easing of restrictions had been introduced. Harsh lockdowns had caused enormous damage to the Israeli economy. During lockdowns, the gross national product decreased by 30 percent, unemployment reached 15 percent, about 26 percent of businesses completely stopped their activities, and 45 percent reduced their activities. 

The survey included an informed consent form. Fifteen BA communications students were instructed to list 30 forums and Facebook groups based on various themes (e.g., politics, culture, family, art, literature, tourism, health, sport, hobbies, etc.). The initial list included 450 forums/groups and the final list included 380 unique forums/groups after the authors and research assistant had deleted duplicates. Each student was given a list of forums/groups to manage, and asked potential interviewees to take the survey in a post. We used the snowball sampling method and, after reaching 70% of the pre-planned sample size, we changed the invitation letter to invite only respondents with specific characteristics (sex assigned at birth, locality, age, and religion) to participate. The final sample of 866 respondents had a 70% response rate. We used quotas for age, gender, and locality of internet users from the sample of the National Social Survey of the Central Statistical Bureau of Israel. Respondents’ email addresses and phone numbers were not included.

The questionnaire included 40 questions on the frequency of exposure to good/bad economic digital news during the pandemic, attitudes toward Israeli Palestinians and English-speaking immigrants, economic and health threats, being harmed economically during the COVID-19 pandemic, health impairment during the COVID-19 pandemic, and demographic characteristics. 

The study was approved by the ethics committee of the Ruppin Academic Center (IRB Approval No.160). The data that support the findings of this study are available on request from the corresponding author. No potential competing interest was reported by the authors. 

### 2.2. Sample

This study was based on an online survey of 866 Israeli Jews. The sex distribution was almost equal, with women comprising 48% of the sample. The mean age was 35.6 years (*SD* = 12.4). Only 3.0 percent of the sample had less than a high-school education, 18.4% had graduated from high school, 9.1% had a vocational studies diploma, 23.0% were enrolled in a bachelor’s degree program, 31.6% held a bachelor’s degree, and 14.9% had a master’s or doctoral degree. The sample’s mean income was *μ* = NIS 8147 (NIS is the Israeli currency; USD 1 = NIS 3.5 at the time of the survey) (*SD* = 5644). 

### 2.3. Measures

The study examined exposure to news in three kinds of media: newspapers, including online versions; internet news feeds; and internet forums, blogs, or social media. As was mentioned in the Section 1, determining whether a news item is ‘good’ or ‘bad’ depends on personal preferences, which makes the assessment extremely complex. The same news item may be perceived as ‘good’ by one individual, but as ‘bad’ news by another. Therefore, we did not use concrete examples of news, but focused on respondents’ subjective perceptions as to whether the news they had consumed was good or bad. The frequency of exposure to economic news was measured based on three questions: “How often are you exposed to bad/good news about the state of the national economy during the COVID pandemic in …” (the three above-mentioned kinds of media), on a scale of 1 to 7, where 1 = “not at all” and 7 = “every day”. Three similar questions were formulated about news on the economic situation abroad. We created four constructs of the frequency of exposure to bad/good news about the economic situation in the national economy/abroad; the internal reliability values (Cronbach alpha) of these constructs ranged from 0.84 to 0.88 (see Table 1).

In order to measure attitudes toward IPs and ESIs who had immigrated to Israel in the last ten years, the General Evaluation Scale [70] was used. The constructs consisted of six items formulated as “When you think about … (population group) who currently live in Israel, how do you feel toward them?” The answers were negative feelings vs. positive feelings; hostility vs. friendliness; suspicion vs. trust; contempt vs. respect; social distance vs. social closeness; and aversion vs. appreciation. These scaled 1–7, where higher values indicated more positive attitudes. The Cronbach alpha values were 0.95 for ESIs and IPs.

Economic threat was measured using two items: “To what extent does the COVID pandemic affect: the economic situation in Israel’s national economy; unemployment in Israel’s national economy”. Health threat was measured using two items: “To what extent does the COVID pandemic affect: your personal health; the health of the entire population in Israel”. These questions were on a scale of 1 to 5. The Cronbach alpha values were 0.86 for economic threat and 0.67 for health threat.

Health impairment during the COVID-19 pandemic was measured using the question: “Which of the following applies to you: 1 = You did not contract Corona; 2 = You were diagnosed as positive for Corona and were asymptomatic; 3 = You contracted Corona and your condition was defined as mild; 4 = You contracted Corona and your condition was defined as moderate; 5 = You contracted Corona and were hospitalized”. Being harmed economically during the COVID-19 pandemic was a dichotomous variable. 

The definitions and descriptions of the variables are presented in Table 1. The reliabilities of all index variables are justified by Cronbach alpha values and factor loadings in SEM (see Section 3).

## 3. Results

### 3.1. Descriptive Statistics

Respondents reported being exposed more often to bad news about the economic situation during the COVID-19 pandemic than to good news (*μ* = 6.30 vs. *μ* = 4.59, respectively, for the national economy, and *μ* = 5.14 vs. *μ* = 4.35, respectively, for the economy abroad, on a scale of 1–7). The frequency of exposure to bad news about the situation of the national economy was higher than for the economic situation abroad (*μ* = 6.30 vs. *μ* = 5.14, respectively), and the frequency of exposure to good news was only slightly higher for the national economy than for the situation abroad (*μ* = 4.59 vs. *μ* = 4.35, respectively). The frequency of exposure to either bad or good news in the national economy or abroad was similar for different kinds of media: newspapers, including their online versions; internet news feeds; or internet forums, blogs, or social media (Table 1).

Attitudes toward ESIs were much more positive than attitudes toward IPs (*μ* = 5.49 vs. *μ* = 3.79, respectively, on a scale of 1–7). Respondents reported high levels of economic threat (*μ* = 4.26 on a scale of 1–5), but a surprisingly moderate level of health threat (*μ* = 3.21). Almost half (43 percent) of the respondents reported having been harmed economically during the COVID-19 pandemic. However, health impairment during the COVID-19 pandemic was relatively low (*μ* = 1.37 on a scale of 1–5).

### 3.2. Effects of Exposure to News about the Economic Situation in the National Economy and Abroad on Attitudes toward Local Minorities

The hypotheses were examined using a structural equation model (SEM) based on the conceptual model of the study (Figure 1). The indices of the model had high goodness of fit: *χ^2^* = 1131.094 (*p* < 0.001), *CFI* = 0.967, *TLI* = 0.960, *IFI* = 0.967, *RMSEA* = 0.041). The results of the SEM assessment are presented in Table 2. 

The study found a positive relationship between the frequency of exposure to bad news about the economic situation in the national economy and economic threat (*β* = 0.450, *p* < 0.001), and a negative relationship between the frequency of exposure to *good* news about the economic situation in the national economy and economic threat (*β* = −0.119, *p* = 0.005). Thus, hypotheses H1.1 and H2.1 were supported. We did not find significant relationships between the frequency of exposure to either bad or good news about the economic situation abroad and the economic threat (*β* = −0.062, *p* = 0.137 and *β* = 0.003, *p* = 0.950 correspondingly); thus, hypotheses H1.2 and H2.2 were not supported.

We found that economic threat was associated with more positive attitudes towards ESIs (*β* = 0.237, *p* < 0.001), but no relationship was found between economic threat and attitudes toward IPs (*β* = −0.029, *p* = 0.551). Hypotheses H3.1 and H3.2 were not supported.

The study found a positive direct relationship between the frequency of exposure to bad news about the economic situation in the national economy and more positive attitudes toward ESIs (*β* = 0.193, *p* < 0.001). We also found a positive relationship between the frequency of exposure to bad news about the economic situation in the national economy and more positive attitudes toward IPs (*β* = 0.109 *, *p* = 0.024). No significant relationships were found between exposure to bad news about the economic situation abroad and attitudes toward either ESIs or IPs. Further, no significant relationships were found between exposure to good news about the economic situation in the national economy or abroad and attitudes toward ESIs or IPs. 

The study also found that a higher respondent income was associated with more positive attitudes toward minorities (*β* = 0.081, *p* = 0.027 for attitudes toward ESIs and *β* = 0.141, *p* < 0.001 for IPs). Respondents who reported higher health impairment during the COVID-19 pandemic reported more positive attitudes toward ESIs (*β* = 0.102, *p* = 0.004), but no significant relationship was found between health impairment and attitudes toward IPs (*β* = −0.038, *p* = 0.310). The effects of age, gender, education, health threat, and being harmed economically on attitudes toward ESIs and IPs were non-significant.

The direct relationships revealed between the frequency of exposure to bad and good news about the economic situation in the national economy, economic threat, and attitudes toward ESIs might indicate mediation. To examine this, we evaluated the indirect and total effects of frequency of exposure to bad and good news about the economic situation in the national economy on attitudes toward ESI. For bad news, the indirect effect was positive and significant (*β* = 0.107, *p* = 0.003), and the total effect was also positive and significant (*β* = 0.300, *p* < 0.001). Thus, economic threat mediates the relationship between exposure to bad news about the economic situation in the national economy and attitudes toward ESIs. The indirect effect was also significant for good news (*β* = −0.026, *p* = 0.003), but the total effect was non-significant (*β* = −0.071, *p* = 0.058). Since there was no significant relationship between the frequency of exposure to good news about the economic situation in the national economy and attitudes toward ESIs, we cannot conclude mediation. 

## 4. Discussion

This study investigated how digital news consumption about the domestic and foreign economic situation in times of crisis is associated with attitudes toward minority outgroups. The contribution of this study is four-fold. First, it contributed to the theory by explaining how the consumption of bad and good economic digital news shapes attitudes toward minorities in times of crisis. Second, it took into account news coverage of the economic consequences of the COVID-19 pandemic in both the national economy and abroad, contributing to the media domestication literature. Third, the mechanism of how economic digital news is related to attitudes toward minorities was explained. Fourth, the study considered the attitudes of the majority population not only toward a disadvantaged minority, which is widely discussed in the literature, but also toward a non-disadvantaged minority, which is usually under-investigated in the literature. 

The study found that the more often respondents were exposed to bad digital news about the economic situation in the national economy, the higher the level of economic threat they perceived. Conversely, when they were exposed to good digital news about the economic situation in the national economy, they perceived a lower level of economic threat. However, we did not find evidence of the domestication effect of global news about the economic situation on Israeli media audiences. One possible explanation is that in times of crisis, when local news is dramatic, people focus mostly on local rather than on global events, and in the respondents’ eyes the importance of local news subordinates the importance of global news. 

According to ethnic competition theory, a higher economic threat is associated with worse relationships between ethnic groups since these groups compete for limited resources [59]. Such competition may be even more pronounced in times of crisis. However, our findings did not support this claim. Firstly, we found that economic threat was not associated with less positive attitudes toward IPs, a disadvantaged minority group. Secondly, the higher the level of economic threat reported by respondents, the more positive their attitudes were toward ESIs, a non-disadvantaged minority group whose economic and social positions were not lower than those of the native population. One explanation for this may be provided in the frame of the threat–benefit model (TBM), according to which immigrants can be perceived not only as a burden to the host society, competing for limited resources, but also as a group that may benefit the national economy [71,72]. Economic units usually seek vital resources, some of which are accumulated and controlled by ethnic minorities [73]. Organizations and businesses employ ethnic minorities to obtain critical diverse resources, such as minorities’ particular knowledge, abilities, and cultural perspectives, which are crucial for firms working with foreign customers [74] or for transnational businesses [75]. Thus, ethnic minorities may be perceived as providing additional resources for economic development, which may be critical for survival in times of crisis. It seems that in times of global crisis, minorities are perceived by the majority as a complementary group, which may contribute to the national economy, rather than as a competing group. 

The fact that the effect of threat on attitudes toward ESIs was positive, and the lack of effect on attitudes toward IPs, may be explained by the prolonged tensions between IPs and the Jewish majority, which are mostly not economic. Conflicts can emerge from various sources beyond economic competition, when social, political, and cultural along with economic elements may contribute to intergroup conflicts. ESI’s ethnic and religious closeness to the majority Jewish population may also play a role [76]. 

An alternative explanation for the positive effects of economic threat on attitudes toward ESIs is the development of broader ingroup identity in times of crisis. Crisis threatens the whole national economy and all citizens regardless of their ethnicity and immigrant experience. In the face of collective peril and threat, people have a natural proclivity to seek affinity and proximity, convey mutual support, and interact collaboratively, suggesting that forming alliances under threat may be one viable collective reaction to pandemics [77]. It seems that, in the face of a common economic threat, intergroup strife fades into the background, and the feeling of belonging to a broad group of people that might be hurt by the crisis overcomes the possible harm stemming from intergroup competition. 

According to the social identity perspective, attitudes toward the ingroup are more positive compared to those expressed regarding outgroups, meaning that granting people membership in the broader ingroup would be reflected in more positive attitudes toward them. In other words, to overcome economic insecurity, Israelis choose to identify more strongly with the widest existing self-inclusive category in Israel—Israeli citizens—which includes an immigrant minority (ESIs) and an ethnic minority (IPs). It seems that, in the face of a common crisis, more frequent exposure to bad digital news about the economic situation in the local economy made the majority population rally around the flag, forget about ethnic competition, feel a part of society as a whole, and develop more positive attitudes toward minorities, whether immigrant or ethnic. Thus, even for groups in protracted conflict, such as Israeli Palestinians and the Israeli Jewish majority, the stress may be shifted in times of crisis to the benefits rather than the contrarieties between the groups.

Economic threat was found to be a mediator of the relationship between the frequency of exposure to bad digital news about the economic situation in the national economy and attitudes toward ESIs. Nevertheless, the existence of significant relationships between media consumption of bad digital news about the economic situation in the national economy and attitudes toward minorities, regardless of the existence of mediation, may indicate other potential mediators. Further studies are needed in this field. 

Since the study mostly concentrated on the economic aspects of the crisis, its results may help to understand what happens to attitudes toward minorities in times of crisis, whether they are caused by war, natural disasters, financial crisis, or other factors.

Study limitations and recommendations for further research. The study is not without limitations. It is based on an examination of a cross-sectional study, and neither establishes causation nor disproves participant choice. The research literature suggests using a comprehensive cross-lagged panel approach to examine causality. For such a strategy, a longitudinal research design is recommended [78], which is what we recommend in future research. 

One limitation of this study is the potential for interpretation bias among participants, as the classification of economic news as ‘good’ or ‘bad’ may vary based on individual perspectives and values. Additionally, the contextual dependency of economic news classification poses a challenge, as the perceived positivity or negativity of economic events may fluctuate depending on broader economic conditions, introducing subjectivity into our analysis. Furthermore, the politicization of economic news may influence participants’ interpretations and introduce biases into our findings [44]. For future research, employing diverse methodological approaches such as qualitative interviews or experimental designs could offer insights into how individuals’ personal perspectives and political affiliations influence their interpretation of economic news, thereby mitigating the potential for interpretation bias identified in this study.

It is important to note that the study investigated the attitudes of the majority population toward minorities, but not the attitudes of minorities toward the majority, which were not the focus of this study. The attitudes of minorities toward the majority in times of crisis the effect may be opposite. In crisis conditions, they may feel more disadvantaged and thus perceive competition as stronger. Further studies are needed in this field.

## 5. Conclusions

The findings mentioned above lead to the important conclusion that communication, social, and psychological theories are less applicable to periods of high uncertainty, and should be adjusted for times of global crisis accompanied by economic downturn. First, in contrast to previous studies on the role of the media in shaping attitudes toward minorities, which found that the consumption of bad digital news was associated with more negative attitudes toward minorities, the current study revealed that, during a crisis, exposure to bad digital news worked differently and was associated with more positive attitudes toward minorities. Second, in contrast to the literature on the domestication of foreign news [21,59], we found that, in times of global crisis, the effects of local news were more pronounced. Third, in contrast to ethnic competition theory, we found that feelings of economic threat during the global crisis, accompanied by extraordinary uncertainty and insecurity, inspired higher cohesion between different groups and more positive attitudes toward them. The study provided evidence that, in times of crisis, bad news for the economy brings good news for social solidarity—people tend to rally around the flag; moreover, this phenomenon even occurs between groups engaged in years-long, protracted conflict. 

## Figures and Tables

**Figure 1 behavsci-14-00232-f001:**
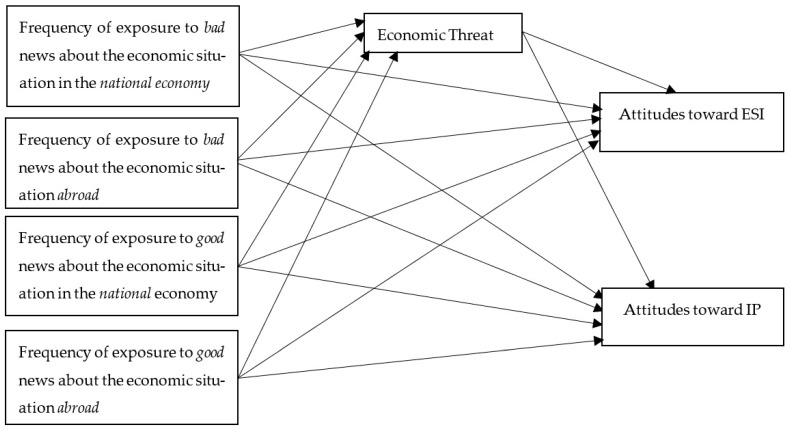
Conceptual model of the study. ESI—English-speaking immigrants. IPs—Israeli Palestinians.

**Table 1 behavsci-14-00232-t001:** Definition and description of variables.

Variables	Cronbach Alpha	Mean	SD
Frequency of exposure to *bad* news about the economic situation in the *national* economy during the COVID pandemic in … (items scaled 1–7)	0.84	6.30	(1.43)
newspapers including online versions		6.14	(1.59)
internet news feeds		6.37	(1.67)
internet forums, blogs, or social media		6.39	(1.67)
Frequency of exposure to *good* news about the economic situation in the *national* economy during the COVID pandemic in … (items scaled 1–7)	0.87	4.59	(1.58)
newspapers including online versions		4.48	(1.69)
internet news feeds		4.59	(1.81)
internet forums, blogs, or social media		4.71	(1.80)
Frequency of exposure to *bad* news about the economic situation *abroad* during the COVID pandemic in … (items scaled 1–7)	0.86	5.14	(1.58)
newspapers including online versions		5.08	(1.66)
internet news feeds		5.16	(1.79)
internet forums, blogs, or social media		5.18	(1.81)
Frequency of exposure to *good* news about the economic situation *abroad* during the COVID pandemic in … (items scaled 1–7)	0.88	4.35	(1.46)
newspapers including online versions		4.30	(1.50)
internet news feeds		4.37	(1.68)
internet forums, blogs, or social media		4.38	(1.70)
Attitudes toward ESIs (items scaled 1–7)	0.95	5.49	(1.31)
negative feelings vs. positive feelings		5.56	(1.46)
hostility vs. friendliness		5.65	(1.46)
suspicion vs. trust		5.43	(1.45)
contempt vs. respect		5.62	(1.45)
social distance vs. social closeness		5.20	(1.50)
aversion vs. appreciation		5.48	(1.44)
Attitudes toward IPs (items scaled 1–7)	0.95	3.79	(1.43)
negative feelings vs. positive feelings		3.96	(1.63)
hostility vs. friendliness		3.94	(1.60)
suspicion vs. trust		3.35	(1.63)
contempt vs. respect		4.36	(1.62)
social distance vs. social closeness		3.37	(1.63)
aversion vs. appreciation		3.76	(1.59)
Economic threat: To what extent the COVID pandemic affects (items scaled 1–5):	0.86	4.26	(0.87)
the economic situation in the Israeli national economy		4.25	(0.94)
unemployment in the Israeli national economy		4.27	(0.93)
Health threat: To what extent the COVID pandemic affects (items scaled 1–5):	0.67	3.21	(0.98)
your personal health		2.70	(1.27)
health of the entire population in Israel		3.73	(0.99)
Income		8147	(5644)
Being harmed economically during the COVID pandemic (scaled 0–1)		0.43	(0.50)
Health impairment during the COVID pandemic (scaled 1–5)		1.37	(0.90)
Age		35.56	(12.44)
Education (scaled 1–6)		4.07	(1.42)

**Table 2 behavsci-14-00232-t002:** Standardized effects of Structural Equation Model.

Relationships	Standardized Effects *β*	*p*
Relationship between the frequency of exposure to bad news about the economic situation in the national economy and economic threat.	0.450 ***	<0.001
Relationship between the frequency of exposure to bad news about the economic situation abroad and economic threat.	−0.062	0.137
Relationship between the frequency of exposure to good news about the economic situation in the national economy and economic threat.	−0.119 **	0.005
Relationship between the frequency of exposure to good news about the economic situation abroad and economic threat.	0.003	0.950
Relationship between economic threat and attitudes toward ESIs.	0.237 ***	<0.001
Relationship between economic threat and attitudes toward IPs.	−0.029	0.551
Relationship between the frequency of exposure to bad news about the economic situation in the national economy and attitudes toward ESIs.	0.193 ***	<0.001
Relationship between the frequency of exposure to bad news about the economic situation abroad and attitudes toward ESIs.	−0.028	0.519
Relationship between the frequency of exposure to bad news about the economic situation in the national economy and attitudes toward IPs.	0.109 *	0.024
Relationship between the frequency of exposure to bad news about the economic situation abroad and attitudes toward IPs.	0.030	0.512
Relationship between the frequency of exposure to good news about the economic situation in the national economy and attitudes toward ESIs.	−0.043	0.340
Relationship between the frequency of exposure to good news about the economic situation abroad and attitudes toward ESIs.	−0.078	0.100
Relationship between the frequency of exposure to good news about the economic situation in the national economy and attitudes toward IPs.	−0.069	0.150
Relationship between the frequency of exposure to good news about the economic situation abroad and attitudes toward IPs.	−0.022	0.665

* *p* < 0.05, ** *p* < 0.01, *** *p* < 0.001.

## Data Availability

Restrictions apply to the availability of these data. Informed concern included the statement that the data would only be accessible to authorized project researchers.

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
