# Peer review of "Attitudes Formation toward Minority Outgroups in Times of Global Crisis—The Role of Good and Bad Digital News Consumption"

_behavsci, 2024, doi:10.3390/bs14030232_

Round 1

Reviewer 1 Report

Comments and Suggestions for Authors

I enjoyed reading this paper. However, I am wondering what "good" or "bad" mean in this study. While the terms, such as "good" or "bad" is very personal. How could the researcher define what it means as good news or bad news, while the news are very political oriented. Please clarify this. 

Other than this, I see this is a good research design. 

Comments on the Quality of English Language

The language is good. Please double check with some minor grammar issues. 

Author Response

Comment 1.

I enjoyed reading this paper. However, I am wondering what "good" or "bad" mean in this study. While the terms, such as "good" or "bad" is very personal. How could the researcher define what it means as good news or bad news, while the news are very political oriented. Please clarify this. 

Answer 1. Thank you very much! In the new version of the paper, we used the definition of ‘good’ and ‘bad’ news based on the operationalization by Kleinnijenhuis et al. (2015), and previous studies in this field. We added a paragraph describing the operationalization of ‘good’ and ‘bad’ based on subjective perceptions of the news consumers.

Comment 2.

Other than this, I see this is a good research design. 

Answer 2. Thank you very much for your appreciation!

Comments on the Quality of English Language

The language is good. Please double check with some minor grammar issues. 

Answer 3. In response to your comment, we once again the paper edited by a professional English academic editor. We hope that after editing, the grammar issues were fixed.

Reviewer 2 Report

Comments and Suggestions for Authors

Lines 87-88: Check if sentence "distinguishing between non-advantaged and disadvantaged minority groups. This research aims to fill this void." is correct because further down (lines 92-94)"Israeli Palestinian citizens (a disadvantaged minority" and "immigrant citizens from English-speaking countries (non-disadvantaged minority group)". What are the correct designations for the two groups under study?

Concepts of intergoup anxiety and realistic threat need clarification on their meaning, as well as bad/good economic digital news. Can the author provide some exmples of the categories mentioned?

Are "bad news" synonym of negative news? Pessimistic views of a particular topic? And the "good news"? More debate on this is needed.

Does the survey contain other variables that help to interpret the results?

Can the differences found between respondednts be attributed to health issues?

Comments on the Quality of English Language

I think the paper is written in a good quality English.

Author Response

Comment 1.

Lines 87-88: Check if sentence "distinguishing between non-advantaged and disadvantaged minority groups. This research aims to fill this void." is correct because further down (lines 92-94)"Israeli Palestinian citizens (a disadvantaged minority" and "immigrant citizens from English-speaking countries (non-disadvantaged minority group)". What are the correct designations for the two groups under study?

Answer 1. Thank You for paying attention! Sure, it should be non-disadvantaged and disadvantaged minority groups. This issue was fixed.

Comment 2.

Concepts of intergoup anxiety and realistic threat need clarification on their meaning, as well as bad/good economic digital news. Can the author provide some exmples of the categories mentioned? Are "bad news" synonym of negative news? Pessimistic views of a particular topic? And the "good news"? More debate on this is needed.

Answer 2. Thank you for this comment! We added the meanings of the realistic threat based on the Intergroup Threat Theory, and expanded an explanation of economic threat as a part of the realistic one.  We also added a definition of intergroup anxiety.

We also provided some examples of ‘good’ and ‘bad’ news: Generally, it is very difficult to describe an economic news item as inherently good or bad because sometimes news is politically oriented, and news evaluation depends on individual preferences. For example, a story about a government that, for example, chooses to dramatically increase welfare support for certain groups (which happened in various countries during COVID) could be framed as good economic news for one group, but perhaps bad news for another group. Based on the ‘good’ and ‘bad’ news operationalization of Kleinnijenhuis et al. [25], and previous studies in this field, we added a paragraph describing the operationalization of positive (‘good’) and negative (‘bad’) news depending on subjective perceptions of the news consumers. News using terms associated with optimism, assurance, enthusiasm, encouragement, relief, grip, rescue, and revival has been used to gauge the positive tone and perceived as ‘good’ news. News using words associated with worry, shock, fear, danger, concern, chaos, tension, nervousness, and anxiety may be regarded as negative (‘bad’) news [26]. Thus, economic news cannot be a priori defined as good or bad; rather, it depends on people’s subjective perceptions of whether the news they had consumed was good or bad. Therefore, in our study we did not use concrete examples of news, but focused on respondents’ subjective perceptions whether the news they had consumed was good or bad. This issue is addressed in the Measures section of the paper. We also mentioned this issue in the Study Limitations.

Comment 3.

Does the survey contain other variables that help to interpret the results?

Answer 3. Yes, the survey also included variables of health threat, being harmed economically during the COVID pandemic, health impairment during the COVID pandemic, income, and demographic characteristics (age, gender, education). All these variables were controlled. The effects of these variables were reported in paragraph 5 of Section 3.2.

Comment 4.

Can the differences found between respondednts be attributed to health issues?

Answer 4. Thank you for this comment! Yes, we also think that health issues can have an effect. To control for it, the model included two variables relating to health issues: health threat and health impairment during the COVID pandemic. We found a positive effect of health impairment during the COVID pandemic on attitudes toward English-speaking immigrants (reported in Section 3.2. of the paper). Effects of health threats on attitudes toward minorities were non-significant.

Comments on the Quality of English Language

I think the paper is written in a good quality English.

Answer. Thank You very much!

Reviewer 3 Report

Comments and Suggestions for Authors

This paper is an important contribution to two areas of current concern: the relationship between the economic impact of historical crisis (in this case: the coronavirus pandemic) and local population's attitude towards immigrants (in this case: Israeli population towards immigrants from English-speaking countries respectively Israeli Palestinians). It is well written and thoroughly documented and the results can be expanded to further dimensions of intersectional crises and their intertwinings.

Please see attached file for some minor improvement suggestions.

Author Response

Thank you very much for your constructive comments!

Comments from the file:

Comment 1. Throughout the paper, please eliminate redundant empty spaces between sentences.

Answer 1. We reviewed the paper and fixed this.

Comment 2. Is the word Anglo communities correct?

Answer 2. We changed it to English communities.

Comment 3. Replace “for example” with “for instance” on p. 4.

Answer 3. We replaced it.

Comment 4. Why this font (p. 5)?

Answer 4. We changed the font for “Dhanani and Franz [56]” (p. 5).

Comment 5. “Mean income was = 8,147 NIS”. What is this? Is this something the readers should know? Maybe the equivalent in USD would be useful.

Answer 5.  We added an explanation: NIS is the Israeli currency; 1 US$ = 3.5 NIS at the time of the survey.